



# An Automated Online Field Instrument to Quantify the Oxidative Potential of Aerosol Particles via Ascorbic Acid Oxidation

Battist Utinger[1], Steven John Campbell[1,2], Nicolas Bukowiecki[1], Alexandre Barth[1], Benjamin
Gfeller[1], Ray Freshwater[3], Hans-Rudolf Ruegg[1], Markus Kalberer[1]

[1]Department of Environmental Sciences, University of Basel, 4056 Basel, Switzerland
[2]Department of Atmospheric and Oceanic Sciences, University of California at Los Angeles, Los Angeles, CA
90095-1565, United States of America.
[3]University of Cambridge, Department of Chemistry, Centre for Atmospheric Science, Cambridge CB2 1EW, UK.

*Correspondence to*: Markus Kalberer (markus.kalberer@unibas.ch)

**Abstract.** Large-scale epidemiological studies have consistently shown that exposure to ambient particulate matter
(PM) is responsible for a variety of adverse health effects. However, the specific physical and chemical properties
of particles that are responsible for observed health effects, as well as the underlying mechanisms of particle
toxicity upon exposure, remain largely uncertain. Studies have widely suggested that the oxidative potential (OP)
of aerosol particles is a key metric to quantify particle toxicity. OP is defined as the ability of aerosol particle
components to produce reactive oxidative species (ROS) and deplete antioxidants in vivo. Traditional methods for
measuring OP using acellular assays largely rely on analyzing PM collected in filters offline. This is labor intensive
and involves a substantial time delay between particle collection and OP analysis. It therefore likely underestimates
particle OP, because many reactive chemical components which are contributing to OP are short-lived and
therefore degrade prior to offline analysis. Thus, new techniques are required to provide a robust and rapid
quantification of particle OP, capturing the chemistry of oxidizing and short-lived highly reactive aerosol
components and their concentration dynamics in the atmosphere. To address these measurement shortcomings, we
developed a portable online instrument that directly samples particles into an ascorbic acid-based assay under
physiologically relevant conditions of pH 6.8 and 37 °C, providing continuous accurate OP measurements with a
high time resolution (5 mins). The instrument runs autonomously for up to three days and has a detection limit of
about 5 μg/m$^3$ in an urban environment, which allows the characterization of particle OP even in low-pollution
areas.





## Introduction

Numerous epidemiological studies have linked anthropogenic air pollution to adverse health effects (Laden et al., 2006; Lepeule et al., 2012; Hart et al., 2015). They demonstrate that exposure to elevated levels of ambient aerosol particles is linked to increased hospital admissions and premature death from various diseases such as cancer, respiratory and cardiovascular diseases (Brunekreef and Holgate, 2002; Künzi et al., 2013). The World Health Organization estimates in a recent report (World Health Organisation, 2016) that 1 in 8 deaths worldwide are related to air pollution. Despite compelling epidemiological evidence, the chemical and physical properties of aerosol particles, and the detailed pathways of particle toxicity, that cause these negative health effects are largely unknown (Bates et al., 2019).

Many countries adopted limit values for the total particle mass as an indicator of particle toxicity. However, several studies have shown that composition, for example, elemental black carbon or transition metal levels, are better proxies for particle toxicity than particle mass concentrations alone (Oberdörster et al., 2005; Koike and Kobayashi, 2006; Godri et al., 2010). Moreover, studies have demonstrated that the oxidative stress resulting from exposure to $PM_{2.5}$ could be a key mechanism to explain the health effects observed upon exposure to particles.

Oxidative stress occurs when an imbalance develops in cells and tissues between reactive oxygen species (ROS), and natural antioxidant defense mechanisms. This imbalance could lead to oxidative stress and therefore trigger various biological effects, such as inflammation, alteration of DNA/proteins, cell damage and death ( Prahalad et al., 2001; Baulig et al., 2003; Donaldson et al., 2001; Li et al., 2003; Offer et al., 2022).

ROS (i.e., inorganic and organic radicals and peroxides such as hydroxyl radical ($\cdot$OH), superoxide ($O_2^-$), hydrogen peroxides ($H_2O_2$), in some cases including organic peroxides) can be delivered exogenously by inhalation of PM into the lung. ROS can also be generated *in vivo* through redox chemistry initiated by aerosol components such as redox-active transition metals and quinones. This ability of PM components to produce ROS, possibly catalytically, through redox chemistry and subsequent antioxidant depletion in biological cells or tissue is defined as oxidative potential (OP) (Bates et al., 2019). Measurement methods have been developed to quantify OP using cellular assays (Lehman et al., 2016) or acellular assays (Campbell, 2021). In epidemiological studies, cellular and acellular assays have shown a correlation between the overall oxidative capacity of PM and its negative effects on human health ( Steenhof et al., 2011; Forouzanfar et al., 2015; Yang et al., 2016; Bates et al., 2019; Zhang et al., 2021). In recent years, several acellular assays have been developed. The advantages of acellular assays are that they are usually cheaper and less time-consuming. The most commonly used assays are the dithiothreitol (DTT), ascorbic acid (AA), glutathione (GHS), and 2,7-dichlorofluorescin/Horseradish Peroxidase (DCFH) assay.

Traditionally, acelluar ROS and OP measurement methods are based on the collection of PM filters, with quantification of OP taking place days, weeks or even months later (Venkatachari et al., 2005; Godri et al., 2011; Salana et al., 2021; Zhang et al., 2021). This results most likely in an underestimation of PM OP because many highly reactive aerosol components, including particle-bound ROS (e.g. ROOH, R$\cdot$, RO$\cdot_x$ species in particular) are short-lived and unstable (Fuller et al., 2014; Zhang et al., 2021). In a recent study we showed that only a very small fraction (<10%) of particle-bound ROS in organic aerosol collected on filters is stable when compared to in situ online DCFH measurements (Zhang et al., 2021), emphasizing the need for the rapid collection of particles to capture the chemistry of highly reactive aerosol components. Other studies, which measured the decay rate of organic radicals (Campbell et al., 2019b), peroxy pinic acid (Steimer et al., 2018), or hydrolysis of organic hydroperoxides in particles (Zhao et al., 2018), also show half-live time from minutes to hours of these unstable compounds.



Therefore, new direct online measurement methods are required with direct-to-assay particle collection and rapid OP quantification to accurately assess the influence of highly reactive components on total OP. Providing robust OP measurements is a key step in achieving a more realistic assessment of the link between particle OP and particle toxicity (Fuller et al., 2014). Thus, various attempts have been made to build online methods and instruments to obtain faster measurements with a higher temporal resolution ( Wang et al., 2011; Fang et al., 2015; Wragg et al., 2016; Gao et al., 2017).

The OP measurement depends not only on the aerosol composition, but also on the assay used. The human lung lining layer contains, in addition to AA, other components such as glutathione, uric acid, and many more anti-oxidants. Calas et al., 2017 (Calas et al., 2017) showed that a simulated/epithelium lung lining to mimic the lung conditions might be favorable. However, Pietrogrande et al., 2019 (Pietrogrande et al., 2019) demonstrated that when assessing the OP with a mixture of antioxidants, the absolute signal of each antioxidant is lower, because the OP is reduced by different antioxidants, which results in higher detection limits.

In this study we present the development, characterization, and first field deployment of a new online instrument that continuously quantifies OP in aerosol particles based on AA oxidation with a high time resolution (ca. 5 min). The online oxidative potential ascorbic acid instrument (OOPAAI) is the first instrument, that provides rapid, highly time resolved OP quantification based on the oxidation of ascorbic acid. We run the OOPAAI using an AA-only based assay to achieve as low a detection limit as possible, allowing quantification of OP in a range of polluted and moderate-pollution environments. The OOPAAI is an instrument, with vastly improved performance, e.g., better detection limit and functionality compared to the method presented by Campbell et al., 2019 (Campbell et al., 2019a), now running at physiologically relevant conditions, and including improved hardware components compared to the instrument developed by Wragg et al., 2016 (Wragg et al., 2016).

## 1    Methods

### 1.1    Reagents

All chemicals were obtained from Sigma-Aldrich and all gases from Carbagas and were used without further purification unless otherwise indicated: ascorbic acid (AA, 99.0%), Dehydroascorbic acid (DHA, 99.0%), 0.1 M HCl solution, 0.1 M NaOH solution, Chelex 100 sodium form, $CuSO_4$ (99.0%), $FeSO_4$ (99%), $Fe_2(SO_4)_3$ (98%), o-phenylenediamine (OPDA, ≥99.5%), HEPES (4-(2-hydroxyethyl)-1-piperazineethanesulfonic acid, ≥99%), α-pinene (≥98%), β-pinene (≥98%), naphthalene (≥99%), zero grade air (Medipac 2000 Superplus, Donaldson Company), $N_2$ gas (purity 99. 999%). The $H_2O$ used to prepare the solutions was purified with an ultrapure water (resistivity ≥ 18.2 MΩ cm$^{-1}$, Synergy, Merck).

### 1.2    Chemical Preparation

All solutions were made fresh every day, unless otherwise specified. Ultrapure water was additionally purified using a fritted column filled with 100 g of chelex 100. 500 ml ultrapure water was added and the valve was adjusted to a flow of one drop per minute. The chelex resin treatment was used to remove trace metals (e.g. copper and iron) from ultrapure water and ensure a stable, low transition-metal-free background, which would otherwise interfere with AA oxidation from the sample. The AA solution (200 μM) was prepared at least an hour before the experiment so that the background drift could stabilize with a 20 mM HEPES buffer at pH 6.8. The HEPES buffer working solution was prepared using 1:10 dilution with chelexed ultrapure water from a stock solution, which is

stored in the refrigerator. OPDA solutions were prepared by dissolving 0.432 g of OPDA in 250 ml of 0.1 hydrochloric acid (20 mM OPDA) immediately before the experiment to reduce OPDA oxidation. AA solutions were stored in opaque plastic bottles (250 ml Nalgene, Merck) which were rinsed with hydrochloric acid to remove trace amounts of metals.

### 1.3    Ascorbic acid chemistry

An AA assay was used to quantify the OP of a given sample by measuring the oxidized form of AA, dehydroascorbic acid (DHA), as shown in Figure 1. The pKa of AA is around 4.1 and therefore at pH 6.8 in an aqueous environment, AA is present mostly as the ascorbate anion ($AH^-$). AA and $AH^-$ can be directly oxidized by metals such as Fe(III) or Cu(II) to DHA, or they can be oxidized over the intermediate monodehydroascorbate radical ($A^-$). In Figure 1 only the most important pathways that occur at a physiologically relevant pH 6.8 are

shown.  In the oxidation of AA, DHA is the first stable reaction product, and therefore DHA was chosen here as a direct measure of AA oxidation. DHA was then further reacted in a condensation reaction with *o*-phenylenediamine (OPDA) under acidic conditions at pH 2 to form the fluorophore (1,2-dihydroxyethyl)-fluoro[3,4-b]quinoxaline-1-one (DFQ) that is quantified in this assay. The chemistry of the AA assay and the mechanism of AA oxidation are described in more detail by Campbell et al., 2019 (Campbell et al., 2019a) and

Shen et al., 2021 (Shen et al., 2021).

The fluorescence detection approach (i.e. reaction of DHA with OPDA) was selected over a direct UV-vis absorbance measurement of AA because fluorescence spectra are often more specific for an individual compound than UV-vis absorbance although the fluoresce detection approach needs an additional condensation step (see Figure S1). AA has a broad UV absorption peak at a wavelength of 265 nm. Many other organic compounds

present in aerosol extracts also have strong absorbance in near-UV (see Figure S2) (Birdwell and Engel, 2010; Huang et al., 2018). Therefore, it is not feasible to device an instrument for continuous online operation using absorbance detection where a decrease in AA would need to be quantified in the presence of a larger background that potentially varies substantially with changes in the organic aerosol composition.





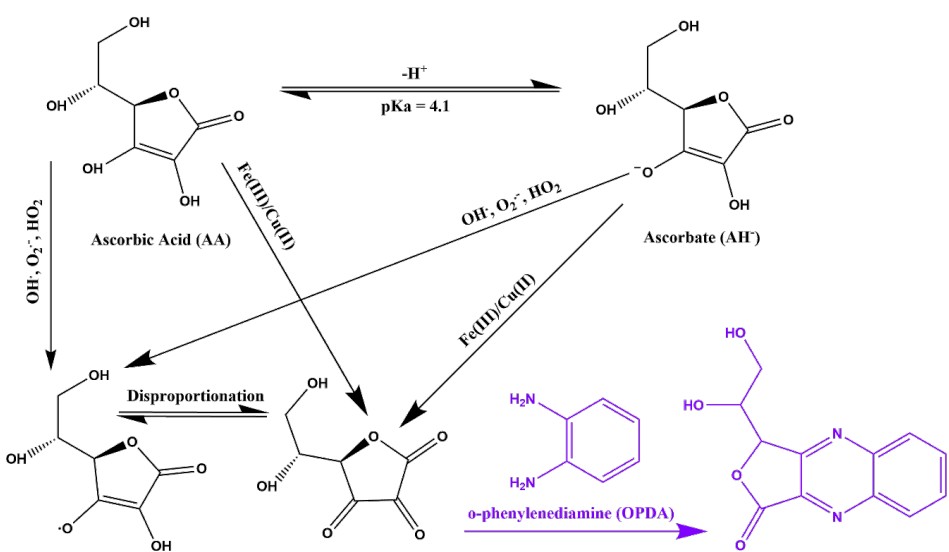


**Monodehydroascorbate Radical (Aˉ)   Dehydroascorbic Acid (DHA)       3-(1,2-dihydroxyethyl)furo-[3,4-b]quinoxaline-1-one (DFQ)**

**Figure 1 - Simplified reaction scheme describing AA and AH- oxidation by ROS and redox-active transition metals (black), and the condensation reaction with o-phenylendiamine (OPDA) to the fluorophore (1,2-dihydroxyethyl)-fluoro[3,4-b]quinoxaline-1-one (DFQ) (purple).**

**1.4    Offline Measurements**

Secondary organic aerosol (SOA) particles (using α-pinene as the precursor) were collected on filters for offline AA analysis to determine particle OP decay. SOA produced from OH-initiated photooxidation of α-pinene is commonly used as an atmospherically representative SOA (Campbell et al., 2019b). High mass concentration SOA (approximately 80 mg/m³) was produced using the Organic Coating Unit (OCU) (Keller et al., 2022) collected on

quartz filters for 100 seconds. These very high aerosol concentrations were necessary to minimize the time between particle generation and analysis and therefore reduce the decay of short-lived OP-active aerosol components. Keller et al., 2022, showed that the OCU produces SOA, even at high masses, with an atmospherically relevant, realistic chemical composition when compared to conditions in the low µg/m³ particle mass range (Keller et al., 2022). In the OCU, an organic precursor is oxidized in a small flow chamber by OH and ozone under controlled

relative humidity (65%) and temperature conditions (fully described by Keller et al., 2022) (Keller et al., 2022). After the OCU, an additional flow tube with additional ozone flow was added to oxidize the VOC that was not oxidized in the OCU and to increase the particle mass. Furthermore, the aerosol passed through a charcoal denuder to remove unreacted ozone and VOCs with an efficiency of 99.9% (Campbell et al., 2019b).

The filters were then collected and extracted after different time intervals and analyzed with an offline AA assay,

using the same chemical conditions as the OOPAAI AA-based assay. A 47 mm quartz filter was extracted directly in 3 ml of 200 µM AA, buffered with 20 mM HEPES, and then vortex for 3 min. The slurry is then filtered with a syringe filter (PTFE), pore size 0.45 µm, Agilent). 900 µL of the extract was then incubated for 20 min in a 37 ° C heating bath, which corresponds to the reaction time between aerosol extract and AA in the online instrument (see below). Subsequently, 100 µl OPDA dissolved in 0.1 M HCl was added and reacted for an additional 3 min



at room temperature, and then the fluorescent signal was measured immediately with a spectrometer (QePro-UV-VIS, Ocean Insight). The fluorescent product DFQ was excited at $\lambda_{ex}$=365 nm, with peak fluorescence emission monitored at $\lambda_{em}$=430 nm.

### 1.5    The Online Oxidative Potential Ascorbic Acid Instrument (OOPAAI)

Figure 2 shows a schematic overview of the OOPAAI. The key components of the instrument are: (1) the PILS (PILS, model 4001, Brechtel) where the aerosol particles are collected into the liquid phase; (2) the heating bath where ROS and the other OP-active particle components react with AA to form DHA under physiological conditions (pH 6.8 and 37 °C) and where the condensation reaction occurs between DHA and OPDA to form the fluorescent product DFQ; (3) the detection cell where the fluorophore DFQ is quantified using a spectrometer.

The aerosol is pumped into the instrument using a vacuum pump (N035.1.2AN.18, KNF) at a flow rate of 16 l/min through a sample inlet (Figure 2, black lines) where a solenoid valve allows switching between a HEPA filter (HEPA-CAP 75, Whatman), which removes 99.9% of particles (to allow for blank measurements with particle-free air), and no HEPA filter. The aerosol then passes through four activated charcoal honeycomb denuders (charcoal honeycomb denuders, Ionicon) with a total length of 14 cm to remove all gaseous oxidizing volatile
compounds that can result in an increased OP signal. After the charcoal denuder, particles > 2.5 μm are removed with a round jet impactor. The impactor is part of the PILS, which was modified for our purposes. The wash flow in the PILS is not operated by the internal peristaltic pump, but with an external peristaltic pump (Ismatec™ REGLO Peristaltic Pump, Fischer Scientific) that can adjust flows for each channel independently. Furthermore, the internal bubble trap was removed from the PILS and replaced with two external ones. External bubble traps
are placed after mixing with the OPDA, contributing to better mixing. To maximize and stabilize the air flow rate to 16 l/min, the internal orifice was enlarged and an additional needle valve was added after the pump. In the condensation chamber of the PILS, the particles are activated by supersaturated water vapor before they reach the impaction plate. Although steam is generated around 100 °C, the measured temperature in the condensation chamber, even close (1 cm) to the tip of the heat generator, is only between 35 °C and 40 °C. Therefore, it is
unlikely that this moderate temperature increase will significantly affect the concentrations of OP-active aerosol components within the PILS. The extraction efficiency of the PILS was determined by comparing the AA oxidation of aerosol measured directly with the PILS (normal setup as described in Figure 2) and aerosol simultaneously collected on a filter (assuming 100% particle collection efficiency on the filter), which was then extracted and the AA oxidation quantified using the fluorescence detection setup in the OOPAAI. Taking the differences of these
two analyses, the collection efficiency of the PILS operated under conditions described here (i.e., very low liquid flow rates and further modifications) was calculated to be 20-25%.

Activated aerosol particles are collected on the quartz impaction plate at the end of the condensation chamber of the PILS (Weber et al., 2010). This quartz impaction plate is washed with a continuous flow of 60 μL/min of AA (200 μM) buffered at pH 6.8 with 10 mM HEPES buffer. HEPES is considered to be a buffer that generally does
not form strong complexes with metals, which is advantageous because the AA assay is very sensitive to redox-active transition metals (see Figure S3) (Ferreira et al., 2015). For several highly OP-active metals such as iron or copper, the speciation of soluble complexes, and the distribution between their soluble form and precipitates, is strongly pH dependent. Therefore, maintaining a physiologically relevant pH is crucial to avoid changing the



soluble fraction of metals and their speciation because the reactivity of AA (or any anti-oxidant) with metals will change significantly depending on the metal speciation and soluble/insoluble fractions.

The buffered AA – aerosol mixture is then pumped from the PILS in a polyether ether ketone (PEEK) tube through a peristaltic pump operated with a PVC peristaltic tubing (AHF, inner diameter 1.02 mm, white-white) at a flow rate of 60 µL/min flow through a cellulose filter (grade 1 filter, Whatman) to remove any insoluble particles. Then, the reaction mixture flows through a heating bath filled with ethylene glycol at 37 °C, with a residence time of 20 min. This ensures that fast-reacting particle components contributing to OP, for example, organic radicals as well as radicals formed by metals catalytically, are captured. Additionally, as DHA is not stable over time (Huelin, 1949) under physiological conditions, an extended residence time would lead to enhanced degradation of DHA, reducing the sensitivity of the OOPAAI. This reaction time of 20 min in the heating bath at body temperature and physiologically representative AA concentrations (200 µM) and physiologically relevant pH (pH 6.8), mimics the conditions in the human lung and the initial reactivity of the aerosol components with AA when aerosol particles get deposited on the lung lining layer.

The liquid flow is then mixed in a t-piece and two helical mixers (1/16 inch, Stamixco) with a 90 µl/min flow of OPDA (20 mM in 0.1 M hydrochloric acid) for approximately 2 min before reaching the detector. The short reaction time was optimized to minimize shifting reaction conditions as a result of the low pH (see SI for details). DHA and OPDA react to the fluorescent product DFQ with a 1:1 stoichiometry, which is then detected by fluorescence spectroscopy. AA and OPDA solutions in the reservoir were continuously degassed with 50 ml/min of $N_2$ to minimize oxidation with oxygen.

All liquid flows in the system are constantly monitored using liquid flow meters (SLF3S-0600F, Sensirion). For fluorescence measurements, the liquid enters the detection cell before being pumped out of the instrument to waste. The detection cell is a home-built unit using a commercial flow through the cell (137-2-40, HELLMA Analytics) with a height of 2 mm, a length of 40 mm and a width of 10 mm. At the end of the detection cell the cuvette narrows where the inlet and outlet are placed perpendicular to the flow direction (see Figure 2). To enhance light transmission through the cuvette, the narrow sides of the cuvette are polished to make them transparent. The LED (Roithner Lasertechnik, type UVLED- 365-330-SMD, peak emission wavelength 365 nm) is soldered on a printed circuit board (PCB), which is screwed to a home-built thermoelectric element to ensure stable temperature conditions and therefore a more stable LED output.

The optical fiber from the LED is in close contact with the polished short side of the detection cell and excites the solute. The emitted light is then collected from the side of the large surface area of the detection cell with a collimating lens (74-UV, Ocean Insight) and focused on a 1500 µm fiber with a numerical aperture of 0.5 (UV-Vis fiber, FP1500RET SMA, Thorlabs). The fluorescence detection cell is enclosed in a light-tight box, which has a mirroring surface (reflecting tape) to maximize the collection efficiency of the fluorescence emission. Right in front of the collimating lens, a bandpass filter (FBH430-10, Thorlabs) from 420 to 440 nm is installed to remove the scattering of the excitation peak. The peak fluorescence emission light at 430 nm is measured using a spectrometer (QePro-UV-VIS, Ocean insight). The fluorescence signal was integrated at over a 10 nm window across the peak emission at 430 nm (425-435 nm). Analog data are collected using a multichannel voltage data logger (1216 series PicoLog, PICO Technologies). A raw spectrum of DFQ, measured with the QePro from Ocean Insight, is shown in the supplementary information (see Figure S4).

The liquid reservoirs, all tubing, the peristaltic pump, the spectrometer, and the flow cell are enclosed in a compartment inside the aluminum box that is insulated and temperature controlled by a thermoelectric cooler (PK

50, ELMEKO) and the corresponding control unit (TPC300, ELMEKO) with two thermistors to measure and regulate the temperature to minimize temperature-related changes in reaction rates (green box in Figure 2). Control of instrument components, as well as data acquisition, is performed using a LabVIEW (National Instruments) script, which sets the input parameters for the peristaltic pump, and monitors the liquid flow rates of the flow meters. A feedback maintains a constant flow at the specified set point. Furthermore, the LabVIEW collects

auxiliary data (various temperatures, and potential errors).

All these components and devices are integrated into an aluminum box (60x60x40 cm) for better thermal stability and for easy transport. A fixed position of the different components ensures improved instrument performance and stability, as movement of key components, such as the optical fibers, has a noticeable effect on the fluorescence signal. The vacuum pump is kept externally to minimize vibrations within the instrument. The waste reservoir is

also kept externally to minimize the amount of liquid inside the OOPAAI.

**Figure 2 - Schematic overview of the OOPAAI. Black lines illustrate air flows where air gets pulled into the PILS via a denuder and an impactor. In the PILS the aerosol is collected with a wash flow containing ascorbic acid (yellow line) and pumped into a heating bath via a grade 1 filter (orange line). OPDA (light brown line) is then added, and the**

**condensation reaction of DHA with OPDA to DFQ takes place (dark brown line). After a mixing piece and two bubble traps, the liquid is pumped into the detection cell, where DFQ is excited by an LED and the fluorescent emission is quantified by a spectrometer. All components are mounted in an aluminum and the components with a light green background are in the thermos-controlled compartment of the OOPAAI.**





## 2 Results and Discussion

### 2.1 OP lifetime of SOA particles

SOA is often a major component of aerosol particles in urban and remote locations (Chen et al., 2022). However, little is known about the OP properties of SOA, and in particular how long-lived the components are which contribute to OP in SOA. Seven filters were collected and stored at room temperature between 2 min and 7 days and analyzed in triplicates. Figure 3 shows the relative decrease in DHA formation over time, normalized to the

online OOPAAI measurement (red square). The offline measurements (black diamonds) and the online measurements are fitted with a two-phase exponential decay function. The stability of OP-active components in α-pinene SOA can be broadly divided into three fractions: a very reactive and short-lived fraction with a half-life of less than a minute, which is only captured by the online OOPAAI, a second fraction that decays slower, and has a half-life time of about 20 hours, and a third fraction that is stable for more than a week. The online signal is more

than three times higher than the immediately measured offline time point, and after a week almost 90% of the OP active components are lost. Overall, this will lead to a significant underestimation of the OP signal when using traditional offline analysis, especially if filter samples are not analyzed immediately. In a recent review Wang et al., 2023 (Wang et al., 2023) reported lifetimes for peroxide from seconds to days. Therefore, the decomposition of these peroxides may contribute to reduced OP on filter samples, as they may well degrade prior to analysis. A

similar decay was also observed for offline measurements for the DCFH assay in several studies (Fuller et al., 2014; Wragg et al., 2016; Zhang et al., 2021). Note that OP decay characteristics could be different for SOA produced from other precursors and in ambient PM due to the presence of other reactive components, such as redox-active transition metals. These experiments clearly demonstrate that OP active components in SOA decay on a time scale (mins to about 1-2 days) that is much faster than typical offline analysis from particles collected

on filters or impactors.

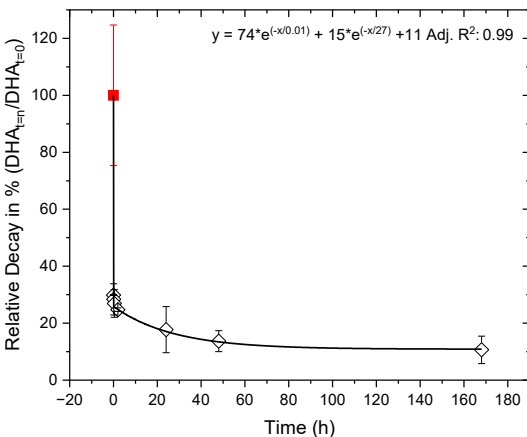

**Figure 3 - Offline decay of AA oxidation by α-pinene aerosol (black diamonds) and comparison with online quantification using the OOPAAI (red square). For the offline decay of OP-active compounds in SOA, DHA concentrations were quantified between 2 min and 7 days after sampling. A two-phase exponential decay function is used for the fit and gives an adjusted R square of 0.998. The relative decay values are normalized to 100% for the online signal. Error bars represent the standard deviation observed over three experimental repeats.**





## 2.2 Response of the OOPAAI: Condensation of DHA with OPDA

For the routine calibration of the OOPAAI, we use aqueous solutions of known concentrations of DHA to determine the instrument response and detection limit with respect to DHA detection. The calibration of the

OOPAAI was performed at pH 2 to assure stability of the solutions during the calibration procedure. Calibration solutions of known DHA concentrations were pumped directly through the PILS without altering anything else in the instrument. The advantages of this calibration method are that there are minimal changes in the instrument setup and at the same time offer an easy and robust method to perform calibrations also during field deployments of the OOPAAI. Figure 4 shows a calibration of DHA in which nine concentrations of DHA ranging from 0-100

µM were measured for 20 min each.

Figure 4A shows the raw detector signal that illustrates the fast time response after changing the DHA calibration solution. At concentrations <10 µM, i.e., the concentration range measured in ambient particles (see Figure 8), the signal equilibrated to a new steady-state DHA concentration within 5 min, which is equal to the instrument's time resolution. This smear-out effect is due to diffusion, wall effects, and turbulence in the tubing, bubble traps, and

the connectors of the instrument. Thus, the instrument is able to resolve changes in the OP concentration under ambient conditions on a time scale of a few minutes. Figure 4B shows the calibration curve for DHA concentrations from 1-100 µM. The error bars indicate the standard deviation obtained for each plateau of the corresponding measurement. This demonstrates a strong linear relationship between DHA concentration and fluorescence signal, with a dynamic range of at least two orders of magnitude. All subsequent calibration curves are blank subtracted

(see first 30 minutes in Figure 4A) and converted from fluorescence counts into DHA concentration using data from the calibration curve (Figure 4) as shown in the first term of Equation 1.

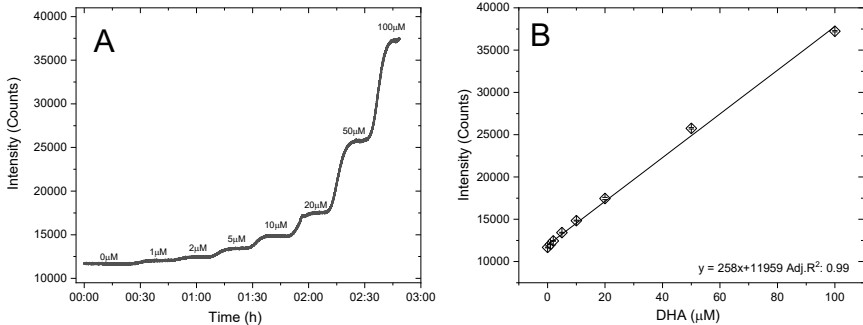

**Figure 4 - DHA calibration curve. (A) Each concentration of DHA solution was measured for 20 min with the setup**
**described in Figure 1. (B) The calibration curve is linear (adj. $R^2 = 0.997$) in the dynamic range of 0-200 µM. The error bars are calculated from the standard deviation of each plateau.**

## 2.3 AA Oxidation by Filter Extracts from SOA, Metals, and Ambient Aerosol

The AA assay is highly sensitive to redox-active transition metals such as Fe and Cu (Shen et al., 2021), but also

reacts with organic components present in secondary organic aerosols, but less efficiently than with metals (Pietrogrande et al., 2022). To determine the sensitivity of the OOPAAI to both metals and organic aerosols, calibrations were performed as illustrated in Figure 5. In Figure 5A, the calibration of α-pinene SOA aqueous

extracts is shown with an adjusted $R^2$ of 0.994 which indicates a good linearity over two orders of magnitude. In Figure 5B, the OOPAAI response for iron (II) sulfate solutions is given with an adjusted $R^2$ of 0.964. The reactivity

of our AA assay is approximately two orders of magnitude higher for iron (II) sulfate than for α-pinene SOA, consistent with Campbell et al., 2019 (Campbell et al., 2019a). However, in ambient aerosol particles, the mass concentration of the organic aerosol is often 1-2 orders of magnitude higher than of redox-active metals ( Kamphus et al., 2010; Calvo et al., 2013; Hama et al., 2018), so despite the lower sensitivity of the AA assay towards organic constituents, they still have a large impact on the overall OP of the particles measured with AA, and therefore

potentially on the oxidative stress that aerosols might generate in the lung.

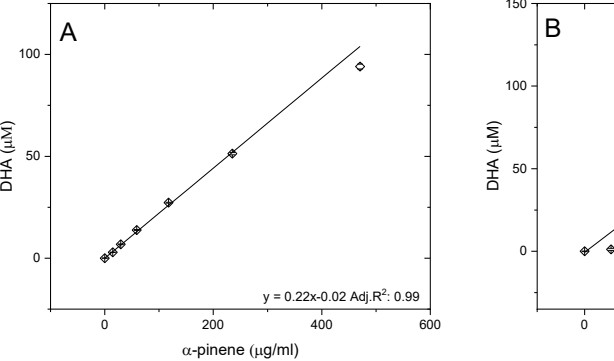
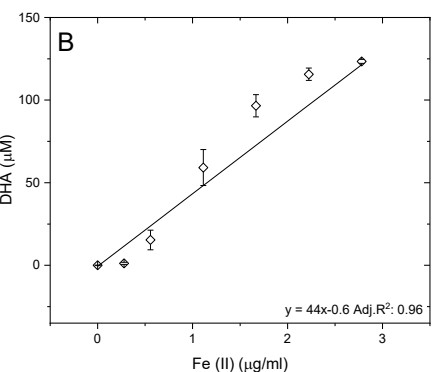

**Figure 5: Response curves generated form (A) α-pinene SOA from filter extracts and (B) from Fe(II)SO₄ solutions. The error bars represent the standard deviation of the spectrometer signal normalized to the DHA concentration during the 30 min measurement. The slope of the linear fits through the data and, therefore, the sensitivity is around 100 times higher for the iron sulphate than for SOA.**


The OOPAAI was also characterized with aqueous extracts of ambient aerosol samples collected on filters. This allowed us to evaluate the instrument performance with solutions of complex chemical composition that include a wide range of organic and inorganic components. For these measurements, the PILS was bypassed and the AA solution was mixed in the instrument with the filter extract using the peristaltic pump of the OOPAAI. Figure 6

shows a series of dilutions of a filter collected in Beijing in 2016 during the Atmospheric Pollution and Human Health campaign (APHH) ( Shi et al., 2019; Campbell et al., 2021). For these measurements, 10 punches (diameter of 1 cm) were extracted in 30 ml of ultrapure chelexed water by vortexing a 50 ml falcon tube for 3 min. Afterwards the filter slurry was filtrated with a syringe filter (0.45 µM pore size) and collected in another falcon tube (full description of the method in Campbell et al. 2019 (Campbell et al., 2019a)). A good linearity ($R^2$ of 0.99) is

obtained, illustrating that complex ambient aerosol extracts can also be quantified with the OOPAAI.

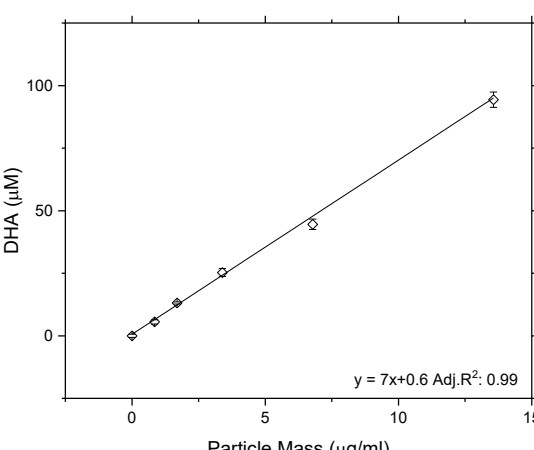

**Figure 6 - Response of the OOPAAI using particle extracts from a filter collected at an urban location in Beijing, China. A clear linear relationship is obtained between the particle mass in the extracts and the instrument signal. Error bars represent variabilty of the OOPAAI signal.**


### 2.4    Calibration with Biogenic and Anthropogenic Aerosol

As a further experiment to show the linearity of the OOPAAI response towards SOA aerosol particles in a wide mass range, the OOPAAI was run in full operational mode (i.e. including the PILS). The aerosol generated by the OCU was diluted to generate particle concentrations in the range of around 50 to 1000 $\mu g/m^3$ and passed through

a home-built charcoal denuder, removing ozone and gaseous VOC residues. In parallel, a scanning mobility particle sizer (SMPS (3080), DMA (3081), CPC (3776), TSI) was operated. The SMPS was set to a scan time of 105 sec and a SOA particle density of 1.4 $g/cm^3$ was assumed (Kostenidou et al., 2007). SOA from two different precursors of VOC, an anthropogenic (naphthalene) and a biogenic (β-pinene) were generated and quantified with the OOPAAI. In Figure 7, the OOPAAI response is shown as a function of SOA mass per cubic meter for the two

SOA types. It illustrates the linear relationship over more than an order of magnitude of particle mass with synthesized, atmospherically relevant aerosol particles, in contrast to the filter extract, which was used for the measurements shown in Figure 5 and Figure 6. The abscissa error in Figure 7 is the standard deviation of the variability in the production of particle mass. The ordinate error represents the standard deviation of the variability of the OOPAAI signal. Naphthalene-SOA particles cause a significantly higher OOPAAI response (i.e. slope) than

β-pinene-SOA, which has also seen by others (Tuet et al., 2017; Chowdhury et al., 2018; Zhang et al., 2021 Offer et al., 2022). This could be because AA reacts very efficiently and catalytically with quinones, which are known oxidation products in naphthalene-SOA, compared with β-pinene-SOA where no quinones are present (Isaacs and Van Eldik, 1997; McWhinney et al., 2013; Roginsky et al., 1999).

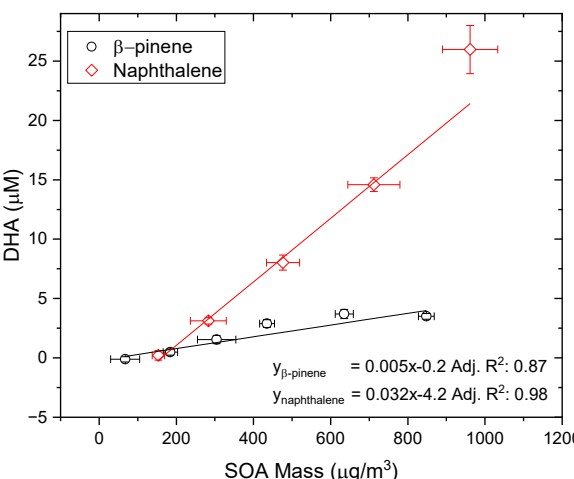

**Figure 7 SOA was produced from an anthropogenic and biogenic precursor, respectively, at different mass concentrations using an organic coating unit (OCU, (Keller et al., 2022)). The OOPAAI signal shows a linear response for both SOA types as a function of particle mass with the anthropogenic SOA being much more reactive towards the**

**2.5     Ambient Measurements**

To determine the response of the OOPAAI for ambient aerosol, proof-of-concept measurements were performed at an urban background location in the city center of Basel, Switzerland. In Figure 8A the raw detector signal from an ambient measurement is shown where OP, in nanomole DHA per $m^3$ air, is plotted over a 4-day period together with $PM_{2.5}$ data for the same time. OP and $PM_{2.5}$ show a clear diurnal cycle with maxima of around 12 nmol
DHA/$m^3$ and 17 µg/$m^3$ respectively, in the evening and during the night. $PM_{2.5}$ concentrations were measured with a Fidas 200 (PALAS) at Basel St. Johannplatz by Luftqualität Nordwestschweiz with a time resolution of a minute (Lufthygine Amt beider Basel). This station is in close proximity (approximately 500 m) to the measurement location where the OOPAAI was operated. The air volume normalized signal of OOPAAI (Figure 8A) has a very similar diurnal trend compared to the aerosol mass, demonstrating that OOPAAI is capable of quantifying OP in
urban particles even at low PM concentrations of only a few µg/$m^3$ with a high time resolution of approximately 5 minutes.

Figure 8B shows the OP data mass normalized for $PM_{2.5}$, with OP concentrations varying more than an order of magnitude from about 0.1 to 2 nmol/µg. The formula used to calculate mass-normalized OP is given in Eq. (1). The first term in Eq. (1) converts the counts of the spectrometer in the OOPAAI into DHA concentrations via the
DHA calibration curve. The second term normalizes the OP signal for volume air through the liquid and air flow rates of the OOPAAI and the third term normalizes for $PM_{2.5}$ mass.

Figure 8B clearly indicates that the OP content in $PM_{2.5}$ is not constant on a diurnal time scale and that in some cases (e.g. 30th Nov., 6pm) OP concentrations change by more than a factor of 5 within about an hour. This demonstrates that the composition of $PM_{2.5}$ and sources with very different OP content can vary quickly in an

urban environment. Organics and metals, for example, have very different OP (Campbell et al., 2023 in prep), as illustrated in Figure 5, and could contribute to the changes in the OP-active aerosol mass fraction. Alternatively, inorganic ions such as nitrate, sulfate and ammonium are OP inactive but may also exhibit a diurnal profile (Timonen et al., 2014) that would affect the total mass fraction of OP. To better understand how such compositional changes, drive the OP particle mass fractions and potentially particle toxicity, more ambient measurements are

required.

The limit of the detection of the OOPAAI characterized in this study, was calculated using the $3_{\sigma bl}$ methods. The detection limit of the OOPAAI for DHA is $0.7 \pm 0.1$ µM. For the urban ambient aerosol measured in this study the OOPAAI is sensitive enough to show a significant response down to $PM_{2.5}$ mass concentrations of around 5 µg/m$^3$.

$$OP \left(\frac{nmol\ DHA}{\mu g\ Aerosol}\right) = \frac{(Counts - Intercept)}{Slope} DHA\ (\mu M)\ x\ \frac{Liquid\ Flow\ Rate \left(\frac{ml}{min}\right)}{Gas\ Flow\ Rate \left(\frac{m^3}{min}\right)}\ x\ \frac{1}{Aerosol\ Mass \left(\frac{ug}{m^3}\right)} \qquad (1)$$

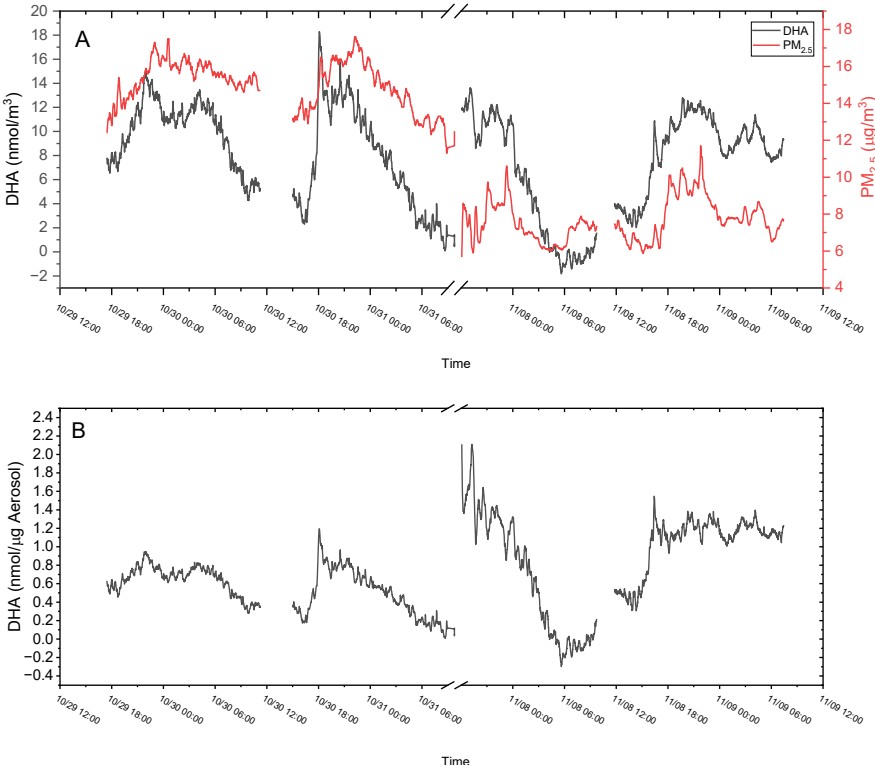

**Figure 8** Ambient OOPAAI measurements in Basel (Switzerland) at Klingelbergstrasse 27, an urban background location in October and November 2022. (A) On the left x-axis, DHA in nanomole normalized to air volume and on the right x-axis $PM_{2.5}$ mass over time is plotted versus time. (B) DHA concentrations normalized to the mass of PM2.5 at

the same time. Gaps in the data are due to changing solutions in the OOPAAI or blank measurements. The LOD for ambient measurements is approximately 5 µg/m$^3$.



## 3   Conclusion

A novel automated online OP instrument has been designed and thoroughly characterized. The OOPAAI can continuously quantify OP concentrations in different types of aerosols with a physiologically relevant AA assay (i.e., pH 6.8 and 37 °C) with an unparalleled time resolution of about 5 min. The OP active components in the particles react within seconds after the particles enter the instrument, and thus very short-lived OP components (the majority of the total OP in SOA particles) are quantified, overcoming limitations of current offline based measurement methods. A comparison of the OOPAAI instrument with offline filter-based OP quantification show that about 90% of OP in SOA particles have a lifetime of minutes to hours and that only about 10% of OP can be detected after a couple of days. Especially the OP components which have the shortest lifetime of less than a minute, are likely formed in the particle via photochemical processes. These compounds may be continuously formed while particles are exposed to sunlight or oxidants in the atmosphere (Alpert et al., 2021) and thus this short-lived OP fraction might also be relevant for health aspects. Therefore, to capture the entire OP concentration, an online instrument as the one presented here with fast collection and analysis time is essential for an accurate quantification of aerosol OP and any offline method is likely to severely underestimate health-relevant OP.

The instrument is portable, capable of running autonomously for about three days, thus is well suited for field campaigns and laboratory experiments. The OOPAAI is sensitive to a wide range of different compounds and aerosol types and can measure their OP over a wide range of environments, from ambient to laboratory-generated particles. Its detection limit for urban aerosol is around 5 $\mu g/m^3$ and a proof-of-principle deployment at an urban background site in Basel, Switzerland, demonstrated the highly dynamic nature of OP active components in $PM_{2.5}$, which can only be captured with the fast-response, high time resolution and robust OP measurement facilitated by the OOPAAI.

This novel online instrument characterizes and accurately quantify OP for the first time in extended field studies at high time resolution, which will be essential for an improved understanding of sources and formation processes of particle OP and to identify the link between particle OP and particle health effects.

### Author contribution

BU developed the instrument, conducted all the experiments, analyzed the data and wrote the manuscript. SJC, AB, BG helped with various experiments. NB wrote the software, RF designed the built many of the electronic components, and HRR assisted with the hardware. MK conceived the study and oversaw the research.

### Competing Interest

The authors declare that they have no conflict of interest.

### Acknowledgements

The authors gratefully acknowledge Sandra Andris-Ogorka and René Glanzmann from the Lufthygieneamt beider Basel for the FIDAS data and the mechanical workshop of the University of Cambridge and Basel for the construction of hardware parts. This work was funded by the Swiss National Science Foundation (grant number 200021_192192/1).





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
