# Peer review of "An Automated Online Field Instrument to Quantify the Oxidative Potential of Aerosol Particles via Ascorbic Acid Oxidation"

_Atmospheric Measurement Techniques, 2023_

## Author Comment (AC1)

**Referee 3**

I found that the authors of this paper designed a thoughtful online field instrument to measure Oxidative Potential (OP) of aerosols. They discussed the chemistry principles and physical characteristics underlying their instrument design. They demonstrated methodologies' functionality with calibration and case studies. At the end, it's revealing to see their method obtained a good correlation of PM2.5 data to support its technological significance. The work is reported in good alignment with the journal's theme. For these reasons I would recommend its publication.

To inspire potential improvement, I have a few questions on a major aspect. The online and off-line OP measurements both had a reaction time of 20 min. Although it's discussed the selectivity of 20min being its optimality toward DHA's stability and relevance to mimicking conditions in lung cells. Can the author discuss how different reaction time will impact the measurement outcome more clearly?

This comment addresses a very important point. Changing the reaction time will favour the sensitivity towards different components. A longer reaction time will shift the sensitivity towards metals that react catalytically and other slowly reacting components. A shorter reaction time will shift the sensitivity towards radicals and other short-lived components. Therefore, by varying the reaction time we could shift the instrument sensitivity for single compounds or classes of compounds. But to compare measurements it is essential to keep the reaction time constant. We identified 20 min as a good compromise between these opposing effects, but we are aware that this is largely an operationally defined number.

Also, Fig 3 suggests the online measurement can capture data within one min; I wonder how does this 1-min reaction be captured by a 20-min residence time?

Maybe there is a misunderstanding. There are three important time constants in the instrument:
(1) The reaction time between OP-active particle components and AA, which is 20 min (line 214). The AA solution is brought into contact with the particles *inside* the PILS, i.e. within seconds after the particles entered the OOPAAI and therefore also very reactive components are quantified, as described in detail in the method section above.
(2) The reaction time of DHA and OPDA, which is 2 min (line 222).
(3) The time resolution with which the instrument can resolve changes in OP content of particles pumped through the instrument. This time constant is about 5 minutes as illustrated in Section 2.5 and Figure 4A, where instantaneous changes of SOA are introduced into the instrument.

The time resolution (5 min) is much shorter than the reaction time of AA with OP-components (20min) because AA is mixed with the aerosol *inside* the PILS, i.e., within seconds after the particles enter the instrument. The 5min time resolution is caused by diffusional broadening during the 20 + 2 min reaction time and transport of the aerosol extract from the PILS to the detection cell (points (2) and (3) above).
In Figure 3, time on the x-axis refers to the time between aerosol generation and sampling. The OOPAAI captures the particles into the AA solution in < 1 minute. Thus, highly reactive species are almost immediately reacted with AA to produce DHA, which significantly more stable, and is subsequently quantified after the 20 min reaction time. The filter sample results shown in Figure 3 were analysed over time to demonstrate the decay of these reactive species. Comparison with the OOPAAI measurements normalised for mass, show that about two third of OP decays on filters, even when analysed a few minutes after collection.

To clarify this, we added a sentence starting at line 274: "In the online instrument, the AA solution is brought into contact with the particles inside the PILS, i.e. within seconds after the particles entered the OOPAAI and therefore, also very reactive components are quantified, as described in detail in the method section above."

In addition to this, I am slightly concerned with the linear fitting on some data sets. In figure 4, it appears to me that the linearity of the intensity-DHA calibration starts deviating after 20 uM, maybe even more at a higher concentration range. Can the author justify the linearity suitable range of the calibration maybe by adding more data points?

We extended the DHA calibration curve beyond the range of 100 µM up to 200 µM the calibration curve slightly starts to deviate from a linear fit. From 1-100uM, however, a linear fit still fits the measured data very well with a $R^2$=99, which we think is a good fit for the measurements shown here. Therefore, we like to keep the calibration curve up to 100 µM in Figure 4.
To address this effect, we added a sentence to the text starting from line 317: "At DHA concentrations larger than 100 µM the calibration curve starts to flatten and such DHA concentrations are therefore considered beyond the linear calibration range of the instrument (data not shown)."

What will be the statistical confidence of the slope and what magnitude of error can be caused? How to handle measurements larger than 100 µM DHA when working with ambient samples?

Confidence intervals are now added to Figure 4. In ambient measurements DHA concentrations were always much lower than 100 µM and thus should not cause a significant uncertainty. If this should indeed occur, the air pumped through the instrument could be diluted.

[Figure]

[Figure]

Another is Figure 5B, where it appears not very appropriate to fit the data by a line anymore- can the authors discuss the implication of the seemingly S-shaped trend?

We added sigmoidal fit to the calibration data in Figure 5B and changed the sentence starting from line 336 to:
"In Figure 5B, the OOPAAI response for iron (II) sulfate solutions is given with an adjusted $R^2$ of 0.99 for a sigmoidal fit."
Furthermore, we added an additional sentence starting from line 337:

"The linear range of the response for Fe(II) is between about 0.5 – 2 µg/ml, above about 2 µg/ml the calibration curve flattens."

[Figure]

With these, I recommend some revision of the work to be formally published.

---

## Author Comment (AC2)

**Referee 1**
This paper describes an instrument developed to measure PM2.5 oxidative potential with a single-component chemical assay, ascorbic acid (AA). The instrument is impressive and the topic well suited for publication in this journal. Some points for the authors to consider:

In the past, the AA assay performed on filter extracts have been conducted in two ways, 1) pure aqueous AA assays in which AA is the only antioxidant and 2) assays in which there are other antioxidants along with AA present, these typically involve monitoring AA in synthetic lung fluid (SLF). The authors chose method 1) since it increases the method detection limit. This is reasonable, but it would be very beneficial if there was some discussion contrasting the reported results of methods using pure AA or AA in SLF, such as what chemical species or sources are associated with AA depletion in each case, what are the similarities and what are the differences, and if possible, what contrasting health effects are observed.

Thank you for pointing that out it is a very important point. We discus in the paragraph starting at line 83 on why we are using only AA (better detection limit) and that mixtures of anti-oxidants like the SLF would likely give different results. Investigating these differences would be beyond this scope of this paper.

The authors find that a-pinene SOA has components that are highly unstable and can only be measured in an AA assay with an online instrument, such as the one described here. The question is, are these species important, ie, toxic? In real lung fluid there may be components that minimize these species and so lessen their toxicity (eg, antioxidants beyond AA) that suppress the ROS transported into by the particles. It has been argued that it is the aerosol particle species that catalytically produce oxidants in vivo that may be the ones that are most important at driving oxidative stress and an inflammation response, such as PAHs and related compounds, and metals, since they can generate ROS without being consumed and can be enhanced in aerosols of mixed chemical components (see more on this below). These compounds also tend to be stable. There is an implicit assumption that all species that react with AA in a pure assay are equally toxic,( eg, see lines 222- 230), maybe actually a filter measurement that only measures the more stable species is a more health relevant measurement? Something to consider.

It is correct that it is currently unknown whether catalytically active components in particles have a more pronounced effect on health or the large number of oxidising organic components such as peroxides or radicals. The OOPAAI has the advantage that it quantified the effect of both, organics and metals.
Importantly, because it is unknown which oxidising particle components are most important for health effects, it is essential to quantify as many as possible and also to quantify components with a short lifetime.

Last line of Abstract, how is the OP LOD units ug/m3?

The OOPAAI can detect the OP of ambient particles at PM concentrations as low as 5ug/m3. This is of course dependent on location, time, and differences in composition, but will give a rough indication that also low pollution levels can be characterised with the instrument.

Line 195-196, why is the PILS collection efficiency so low, 20-25%. It is not clear what liquid flow rates are very low which is stated to be the cause? Please clarify.

We changed the sentence starting on line 195 to the following to clarify it:

"Taking the differences of these two analyses, the collection efficiency of the PILS operated under conditions described here (i.e., very low liquid flow rates for the wash flow in the PILS and further modifications as shown in Figure 2) was calculated to be 20-25%."

Could one add flow rates (air and liquids) to the schematic of Fig 2?

Flow rates have now been added to Figure 2.

Where do the 75 to 80% of the missing particles go? Is the resulting ROS measurement corrected for this low sampling efficiency?

The particles, which are not collected by the PILS are either removed with the air flow or with a small liquid waste flow within the PILS, which assures that air bubbles build up in the OOPAAI liquid flow system.
The data in Figure 3 is corrected for the collection efficiency, because absolute values between online and offline measurements are compared. For all the other measurements the correction is not applied to illustrate the actual measurement capability of the OOPAAI. We added a sentence at line 197 to clarify that:
"The online measurements in Figure 3 are corrected for the lower particles collection efficiency, to ensure inter-comparison between online and offline measurements, but the other measurements are not corrected, because we want to show the actual capability of the instrument at its current development stage."

Line 208, what is the pore size of the cellulose grade 1 filter, which will define the size of insoluble particles that pass through this filter.

We added the pore size of 11 µm on line 212.

Do the authors know if the denuders are necessary? Eg, what possible gases would interfere?

For lab generated SOA, the denuder is necessary, because high ozone concentrations were used to generated SOA and the ozone strongly interfered with the system. Other gases, e.g. $H_2O_2$ or gaseous peroxides, might also cause unwanted interferences. To assure comparable lab and field results, the denuders were used in lab and field measurements.

Do the denuders actually produce more reactive gases?

To the bet of our knowledge, denuders do not produce any reactive gases. We don't see any change in the blank signal when measuring clean air with the denuder or without it. The same is the case if we add a HEPA filter and measure ambient aerosol. We also showed in a previous study that the activated charcoal denuders are very efficient in removing gas phase oxidants such as ozone (~99.9%) (Campbell 2019 ES&T), a gas which can produce ROS such as OH in aqueous media, thus convoluting particle based oxidative activity.

One might wish to compare with other forms of denuders. Why were such a large number of denuders in series used?

Maybe there is a misunderstanding, we only used four honeycomb denuders with a total length of 14 cm. It is a good idea to reduce the total length of the denuder to reduce particle

losses, which even in this case are fairly minimal (10-15%) and therefore improve the sensitivity, but for ambient measurement it is rather challenging to identify the minimum denuder length which can vary depending on ambient conditions (e.g. NOx and ozone concentrations). It is crucial to remove gas phase artefacts which may convolute OP measurements, as our interest is in the OP of ambient particles. Therefore, we added the current number of denuders to our setup to ensure that gas phase background signals are reduced.
Comparing other denuder forms is a good idea but was not the focus of this study.

Is it true that particles deposited in the lung fluid will be at the lung fluid pH? What about if incorporated into specific components of the lung fluid, such as macrophages?
There might be a misunderstanding. Macrophages are cells, which can be present on the inner surface of the lung but are not the same as the lung lining layer. There are several publications summarizing the pH of the lung lining layer like as in Vaughan et al., 2003.

Line 284. The point is not clear. Metals are not SOA. Will metals have the same issue as fresh SOA, ie, if not measured immediately their contribution to OP AA will be under-measured?

This is a misunderstanding. We are aware that metals and SOA are not the same and we have reworded the sentence starting at line 289. It now reads:
"Note that OP decay characteristics could be different for SOA produced from other precursors. In addition, in ambient PM, inorganic particle components such as redox-active transition metals could also contribute to OP."

Line 325-330. The conclusion that the AA assay response is x100 higher than that for a-pinene SOA, but SOA concentrations are much higher than metals so this evens things out is an overly simplistic analysis since the experiments were based on single pure components. Real aerosols are a mixture of many compounds, such as metals and PAH-(and related compounds) that can synergistically affect OP. As just one example, Fenton reactions that produce ROS are enhanced in the presence of semi-quinones that cycle Fe(III) back to Fe(II).

We do not claim that the AA response towards metals and SOA "even out" in the ambient atmosphere. However, we like to point out that in the ambient atmosphere the SOA mass is often much larger than the metal mass. Therefore, one should not disregard the OP contribution of SOA components to the overall particle-induced OP.

Line 341 to 350, the tests with Beijing filters, is not clear. Was the idea that the online analytical system was being compared to a manual analysis, both following the same protocol? It seems like the filter was extracted in a water solution containing AA and then that extract analyzed by the online and manual methods for a comparison.

Thank you for pointing out that this is not clear. The motivation for this experiment was to test the response of the OOPAAI for complex mixture of aerosol (ambient), but with a simplified setup without the aerosol collection unit. We changed the respective sentence starting from line 354 to the following:
"To characterize the OOPAAI with a more complex chemical system we used an aqueous extract of ambient aerosol samples collected on filters."

Fig. 7 caption is cut off.

We added the missing words and the figure caption of Fig. 7 now reads: "The OOPAAI signal shows a linear response for both SOA types as a function of particle mass with the anthropogenic SOA being much more reactive towards the AA assay."

Equation 1 has no correction for PILS collection efficiency; does this mean the measurement is about 75% too low?

We added a sentence at line 197 to clarify that: "The online measurements in Figure 3 are corrected for the lower particles collection efficiency, to ensure inter-comparison between online and offline measurements, but the other measurements are not corrected, because we want to show the actual capability of the instrument at its current development stage."

Line 429 to 431, there is no proof for this statement. As noted above, the authors are assuming that the more unstable components measured with the online AA system, but not the filters are health relevant, but what is the justification? Maybe qualify this by stating something like, assuming all species, stable and unstable are of equal toxicity…. Or add to the line; severely underestimate health-relevant OP for cases of high concentrations of relatively fresh SOA.
All chemical OP measurement techniques and methods are not able to say anything directly about the actual toxicity of their results. We do not claim that we quantify particle toxicity but a potentially useful proxy for particle toxicity.

---

## Author Comment (AC4)

**Referee 2**

The development of an automated online field instrument to quantify the oxidative potential of aerosol via ascorbic acid oxidation is presented in this manuscript. The authors conducted a series of calibrations of aerosols with known sources or compositions as well as reported data from ambient measurements. The use of fluorescence detection is important to avoid background interference from absorbance-based measurements. I think the manuscript is well written and the data is carefully analyzed. I have a few clarifying questions for the authors listed below.

• How did AA maintain freshness for multiple days of field measurements?

To clarify that we changed the sentence starting at line 22 to: "To reduce oxidation and keep the solutions fresh over multiple days, the AA and OPDA solutions in the reservoir were continuously degassed with 50 ml/min of $N_2$. There was only a minimal increase in the blank signal over time, which was detrended by a linear fit if necessary."

• Would the change in AA concentrations in the instrument over time (via natural decay) affect the kinetics of AA-aerosol oxidation reactions?

The decay rate of the AA is relatively slow, and in measurement campaigns is regularly replenished every ~2 days, so the decay of the initial AA concentration is not more than 5%. The AA is also in substantial excess compared to redox-active aerosol components, particularly during ambient sampling. Therefore, we do not expect these small changes in AA to affect OP measurements.

• A limit of detection for urban ambient aerosol measurements was reported to be around 5 µg/m3. Given that different types of aerosol components may have very different reactivities towards AA oxidation, do the authors have any information about what major aerosol composition contributes to the current observation?

This is a very good point, but unfortunately, we don't have this information. This publication is focussing on a detailed characterization of the OOPAAI and a proof-of-principle for an ambient atmosphere application. A full characterization of the ambient aerosol and which ambient particle components contribute to the OOPAAAI signal, would go beyond the scope of this paper.

• How was the slope for ambient measurements determined? (Line 410)

This might be a misunderstanding. The "slope" in line 426 (equation 1) is the slope of the DHA calibration curve (see Fig. 4) and is used to convert counts into DHA concentrations.